# Understanding Enablers and Barriers to the Implementation of Nutrition Standards in Publicly Funded Institutions in Victoria

**DOI:** 10.3390/nu14132628

**Published:** 2022-06-24

**Authors:** Emalie Rosewarne, Wai-Kwan Chislett, Briar McKenzie, Cliona Ni Mhurchu, Tara Boelsen-Robinson, Miranda Blake, Jacqui Webster

**Affiliations:** 1The George Institute for Global Health, University of New South Wales, Sydney 2006, Australia; waikwanchislett@gmail.com (W.-K.C.); bmckenzie@georgeinstitute.org.au (B.M.); c.nimhurchu@auckland.ac.nz (C.N.M.); jwebster@georgeinstitute.org.au (J.W.); 2National Institute for Health Innovation, School of Population Health, University of Auckland, Auckland 1023, New Zealand; 3Global Obesity Centre, Institute for Health Transformation, Deakin University, Geelong 3220, Australia; tara.b@deakin.edu.au (T.B.-R.); miranda.blake@deakin.edu.au (M.B.)

**Keywords:** nutrition standards, food procurement, policy implementation, publicly funded institutions

## Abstract

Effective implementation of nutrition standards in publicly funded institutions can facilitate healthy food and beverage consumption by communities and populations, which can enable improvements in dietary intake and reduce disease burden. This study aimed to understand stakeholder perspectives on the implementation of government nutrition standards in publicly funded institutions in the Australian state of Victoria, as well as to determine enablers and barriers to successful implementation. Pre-interview questionnaires and semi-structured interviews were administered to stakeholders involved in the implementation of nutrition standards in publicly funded institutions in Victoria. The Interactive Systems Framework, which allows understanding of the infrastructure and systems needed to implement policies, was used to design the survey instruments and guide the data analysis. Forty-four stakeholders were interviewed, including program implementers, support personnel and food providers, across public sector hospitals and health services, workplaces, sport and recreation centres and schools. Though translated materials and resources have been developed for end-users to facilitate uptake and implementation, current nutrition standards were perceived to be long and complex, which hindered implementation. The existence of a government-funded implementation support service enabled action by providing technical support, troubleshooting and capacity-building. A specific pathway for successful guideline implementation was determined through the analysis. Opportunities to close the policy-implementation gap were identified. This will be crucial to maximising the impact of nutrition standards on population diets and reducing diet-related disease. Strengthening the guidelines and their governance, streamlining the support system and overcoming barriers within and outside of implementing organisations, are urgently required to propel statewide progress.

## 1. Introduction

Nutrition standards in publicly funded institutions (e.g., schools, workplaces and hospitals) are a key opportunity for creating healthy, supportive food environments [1,2]. Nutrition standards are “food- and/or nutrient-based criteria applied to the procurement, preparation, provision, or sale of foods” [3]. Applying these standards can facilitate healthy food and beverage consumption by communities and populations in publicly funded institutions [4,5]. This should enable improvements in overall diet quality and reduce the burden of disease attributable to unhealthy diets [6,7].

At least one-third of countries have national nutrition standards for one or more publicly funded institutions [3], likely reflecting governments’ ability to regulate foods and drinks purchased with public money and provided in public facilities [1]. However, little is known about the effectiveness of such policies as very few evaluations have been conducted [3]. There is some evidence from high-income countries to suggest that nutrition standards can increase the availability and purchases of healthier foods within publicly funded institutions [8]. However, more research is needed.

In Australia, state and territory governments are responsible for the development and adoption of nutrition standards in publicly funded institutions [9]. There are currently 24 policies with nutrition standards for schools, workplaces, hospitals and other publicly funded institutions across the eight Australian jurisdictions [10]. These policies and nutrition standards vary in design and comprehensiveness between jurisdictions and institution types [3,10]. There have been a few evaluations demonstrating limited compliance with current nutrition standards. The majority of studies have been conducted in schools in specific jurisdictions and the results have been summarised by Wu et al. [11].

Victoria, Australia’s second most populated state of almost 6.7 million people [12], has government nutrition policies spanning a wide range of publicly funded institutions. The Healthy Choices Guidelines (HCG) for health facilities, workplaces, parks and sport and recreation centres [13,14,15,16], were developed in 2010 (last updated in 2020) and are governed by the Department of Health [17]. The guidelines are underpinned by a set of nutrition standards outlined in the “Food and drink classification guide” [18]. The School Canteens and Other Food Services Policy (SCFSP) for schools [19] was developed in 2006 and is governed by the Department of Education and Training [20], and includes nutrition standards. The intention of HCGs and SCFSP (hereafter referred to collectively as “guidelines”, except where otherwise specified) is to increase the supply and promotion of healthy food and drink options and reduce the supply and promotion of unhealthy options in food retail, vending and catering in publicly funded institutions. The Government funds a support service and recognition program to facilitate implementation of the guidelines. The Healthy Eating Advisory Service (HEAS) is a free government-funded support service established in 2012 and delivered by Nutrition Australia (Victorian Division). The purpose of HEAS is to provide information and guidance, tools and resources, implementation support and technical assistance to enable publicly funded institutions to implement the guidelines [18]. HEAS has developed a key tool, FoodChecker, to facilitate menu, vending, product and recipe assessments and provide immediate feedback and advice to create healthier food environments [17]. The Achievement Program (AP) is a recognition program established in 2012 and is delivered by Cancer Council Victoria [21]. Institutions working towards the AP’s “healthy eating and oral health” benchmark can also receive support through the AP to create healthier food environments. These Government-funded services for the implementation of nutrition standards in publicly funded institutions are unique to Victoria in Australia [10].

Pockets of success have been observed. Evaluations of changes in foods and drinks available as a result of full or partial guideline implementation in small samples of Victorian publicly funded institutions have illustrated positive changes to food environments and food purchases [22,23]. For example, 33% of Victorian YMCAs (Young Men’s Christian Association) included in an evaluation had removed all sugar-sweetened beverages from their aquatic and recreation centres in 2015 [22], and implementing the HCGs in vending machines across three hospitals resulted in a 56% decrease in sales of unhealthy drinks [23]. However, there is no statewide data on guideline implementation or evaluation [24], and these are likely to be low based on evidence from across Australian jurisdictions [11,25,26]. Victoria has more than 1500 government schools [27], over 300 hospitals and health services [28], and 79 local government areas responsible for 9500 sport and recreation facilities (although not all provide food and drink) [29]. As such, there is large potential for government nutrition standards in publicly funded institutions in Victoria to have an impact on population diets. Identifying enablers and barriers to the implementation of nutrition standards in publicly funded institutions in Victoria is important in understanding why there is yet to be a full-scale shift towards statewide implementation.

Our previous work has illustrated the scope for improvements to policy design to reduce barriers to implementation and maximise the impact of nutrition standards for publicly funded institutions in Australia and globally [3,10]. This study aimed to understand stakeholder perspectives on the process of implementing government nutrition standards in publicly funded institutions in Victoria and determine enablers and barriers for effective implementation with a view to providing insights into facilitating statewide implementation progress.

## 2. Materials and Methods

Pre-interview questionnaires and semi-structured interviews were administered to stakeholders involved in the implementation of nutrition standards in publicly funded institutions in Victoria. The survey instruments and data analysis were informed by the Interactive Systems Framework, which allows understanding of the infrastructure and systems needed to implement policies [30].

### 2.1. Sample and Recruitment

Potential participants were stakeholders who were planning to be involved, are involved or had been involved in the implementation of the HCG or SCFSP. Participants were stakeholders in publicly funded schools, workplaces, hospitals and health services, and sport and recreation centres in Victoria. They were expected to include local government and state government employees, food providers (e.g., school canteen and food outlet managers, food service providers, manufacturers and suppliers), support service staff, health promotion officers and dietitians, and other staff within publicly funded institutions.

An email invitation to participate was sent to the HEAS mailing list in August 2019 (HCG) and February 2020 (SCFSP). Snowball sampling was used to identify additional potential participants to obtain a wide range of perspectives on implementation, and enablers and barriers. Invitees expressed their interest by emailing the research team, who then sent the participant information and a consent form and confirmed eligibility. Stakeholders agreed to participate by return email, including a signed consent form and agreeing to an interview time. Interviews were conducted from August 2019 to March 2020, either in person or online using Skype for Business or Zoom. Recruitment and interviews of school stakeholders were terminated earlier than planned on 20 March 2020, when schools in Victoria were closed due to COVID-19.

### 2.2. Survey Instruments

There were two components to this study: A pre-interview questionnaire and a semi-structured interview. The questionnaire and interview guide were informed by the Interactive Systems Framework (ISF) for Dissemination and Implementation [30]. The ISF focuses on the infrastructure and systems needed to disseminate and implement policies, accommodating perspectives of funders, researchers, implementers and support providers [30]. Questions centred around the three framework domains: (1) synthesis and translation system (distilling information about the guidelines/policy for end-users), (2) support system (supporting end-users to implement the guidelines/policy) and (3) delivery system (implementation by end-users). Use of this framework will allow synthesis of insights from stakeholders across the policy design and implementation spectrum pertaining to the implementation of nutrition standards for publicly funded institutions to identify opportunities for system improvements that will facilitate statewide progress.

#### 2.2.1. Pre-Interview Questionnaire

A 20-question pre-interview survey was administered to stakeholders at the time of the interview for face-to-face interviews or via email before the interview for online interviews. The tool collected: background information on the organisation, knowledge levels relating to the guidelines/policy, and basic information relating to implementation in the stakeholder’s organisation. The purpose of the questionnaire was to enable the researchers to tailor the semi-structured interview to the participants, describe characteristics of the sample and provide an overview of implementation.

#### 2.2.2. Semi-Structured Interviews

Semi-structured interviews were conducted by two researchers, E.R. and W-K.C. The semi-structured approach allowed interviewers to adapt the questions based on each interviewee’s response to the pre-interview questionnaire (e.g., to omit irrelevant questions) and involvement in implementing the guidelines (e.g., to ask manufacturers/suppliers, support personnel and program implementers questions in a way that was relevant to their role). It also enabled interviewers to ask follow-up questions and therefore enabled further exploration of the topic. Interviews were audio-recorded. Permission to record was obtained from participants via a signed consent form and verbally at the beginning of the interview. The interviews were manually transcribed verbatim by a private transcription company.

### 2.3. Data Analysis

Data analysis was performed by three researchers, E.R., W.-K.C. and B.M., with 10% of transcripts being independently double-coded and compared for consistency. For the double-coded interviews, any disagreements between researchers were resolved through discussion. For the single-coded interviews, any uncertainty in coding by one researcher was discussed with the other two researchers. Participants were characterised according to the type of organisation (e.g., school, workplace, hospital) and their role in guideline implementation (e.g., implementer, support person, manufacturer/supplier, multiple roles). Data from the pre-interview questionnaire were entered into a Microsoft Excel spreadsheet and descriptive statistics were generated. Interview data were imported into NVivo for data management. A combination of deductive and inductive methods was used to thematically analyse the transcripts. Themes relating to the ISF constructs were identified deductively. Additional themes were inductively identified through line-by-line transcript analysis and discussion among the three researchers.

### 2.4. Ethics and Consent

This study was approved by the University of New South Wales Human Research Ethics Advisory Panel (HREAP) G: Health, Medical, Community and Social (HC190242) and was performed as per the Declaration of Helsinki. Written informed consent was obtained from all participants before the interview.

## 3. Results

Themes and constructs identified in this study are illustrated in the Adapted Interactive Systems Framework (Table 1).

### 3.1. Participants

Forty-four stakeholders from 36 organisations were interviewed. Interviewees were program implementers (*n* = 28), support personnel (*n* = 7), both implementers and supporters (*n* = 4) and manufacturers/suppliers (*n* = 5) across hospitals and health services, workplaces, sport and recreation, and schools (Table 2). Thirty-one interviews were conducted about the implementation of the HCGs and 13 interviews were conducted about the implementation of the SCFSP. Themes relating to the implementation of both the HCG and SCFSP were similar, so the results are presented together. The average duration was 48 min (range: 22 to 79 min).

### 3.2. An Overview of Implementation: The Pre-Interview Questionnaire

Forty-one interviewees completed the pre-interview questionnaire. The questionnaire was not appropriate for the other three participants who were from the state-funded support service and were involved in implementation across many different organisations.

The size of organisations implementing the guidelines ranged from 55 staff (workplace, local government area) to approximately 77,000 staff and students (workplace, university). The majority of interviewees stated that health was a high priority within their organisation (*n* = 25) although healthy eating was less of a priority (*n* = 29). Most participants were in the process of implementing the guidelines (*n* = 19) and a smaller proportion had fully implemented the guidelines (*n* = 9). Participants were aware of, and received support from, government-funded services including HEAS (*n* = 41 and *n* = 35, respectively) and AP (*n* = 29 and *n* = 16, respectively). Almost all stakeholders (*n* = 38) reported their organisation provided food through more than one method, including catering, fundraising, food outlet/retail and vending machines. Most had assessed the nutritional composition of foods provided (*n* = 37) and made changes to improve foods (*n* = 39). More participants reported experiencing barriers to implementing the guidelines than enablers (*n* = 38 and *n* = 29, respectively). Around one-quarter were monitoring impact (*n* = 11). Just over half of the interviewees cited adequate support (*n* = 27) and resources (*n* = 22) provided by their organisation to implement the guidelines (Appendix A).

### 3.3. Enablers and Barriers to Implementation: The Semi-Structured Interviews

The following sections describe the enablers and barriers to implementation identified by participants. The results are organised by adapted ISF constructs (Table 1).

#### 3.3.1. Synthesis and Translation System

##### Accessibility

Stakeholders shared their experiences of accessing, understanding and applying the guidelines alone or through materials and resources translated for end-users, such as the FoodChecker website. Interviewees stated there was a general lack of awareness that the guidelines exist. Participants’ first exposure to the guidelines was usually through a support service (e.g., HEAS) or by searching online themselves. One stakeholder shared specific challenges in accessing the SCFSP “because it’s named something bizarre like Healthy Canteen Kit” (Support—school) and seven interviewees said that the SCFSP was out-of-date and not relevant, having not been updated in over 10 years.

##### Understandability

The guideline documents were generally described as long and complex. Most interviewees believed that a strong understanding of nutrition was required to understand the guidelines and some reasoned that this was why retailers, manufacturers and suppliers had difficulty interpreting and applying the guidelines. One participant explained the potential impact of this on policy uptake: “it’s not going to be able to be implemented more broadly if you have to put specialist experts into the translation” (Implementer—workplace). However, another stakeholder suggested that the end-user resources that had been developed negated this: “a lot of people now go straight to FoodChecker for things and they don’t actually look at documents like the classification guide or whole policy document… so maybe it’s just directing them to the right things” (Support—sport and recreation).

##### Utility

Perspectives on the utility and application of the guidelines were mixed. Some people perceived them as useful for enabling the creation of healthy eating environments and a valuable tool to advocate for change to foods within organisations. It was felt that FoodChecker was a valuable tool and enabled policy uptake and implementation by facilitating menu assessments and supporting recipe modification, saving organisations time and resources compared to the manual process. Others indicated the process of entering data into the FoodChecker system was still too time-consuming, and some interviewees suggested the tool could be improved by increasing the number of foods and drinks included in the product database.

Application challenges cited by stakeholders included: applying the guidelines across more than one method of food provision (e.g., retail and catering) or multiple site locations; differences in nutrient criteria between the HCG and SCFSP, which participants said made it difficult for manufacturers, retailers and suppliers who were supplying food across all publicly funded institutions; and disparities between the nutrient criteria in Victoria and other state and territory guidelines, which prevented operation across all Australian institutions.

Fifteen interviewees raised specific concerns about the content of the guidelines. The top concern (*n* = 7) was the application of nutrient criteria such that foods and meals could be manipulated to be classified as green or amber. For example, four program implementers did not agree that baked goods (e.g., muffins, cakes and savoury pastries) should be able to be classified as green with a few recipe modifications to reduce harmful nutrients in them. One stakeholder argued that “Cakes aren’t green. They aren’t meant to be made with wholemeal flour and Nuttelex and one egg… Cakes are red, and that’s ok because they are special occasion foods” (Implementer—Workplace (university, government facility)). Some program implementers disagreed with guidance on specific food categories. For example, the focus on reduced-fat dairy products (*n* = 4) was perceived to be out-of-date with current evidence on the topic, in particular the “recent position statement from the Heart Foundation” (Implementer—Hospital or health service).

#### 3.3.2. Support System

##### Structure of the Support System

The research highlighted the fact that the support system is complex with multiple providers, levels of support and pathways to receive support. Figure 1 illustrates the many support pathways identified in the interviews.

##### Roles in Providing Information and Support for Implementation

Government support service providers, HEAS and AP, provided direct support to facilities as well as support through the health promotion workforce. The health promotion workforce comprised health promotion officers in local government and community health organisations,; school nurses and pastoral care staff,; and staff in community organisations, and provided direct support to facilities. However, generally, the health promotion workforce support was only available to organisations if healthy eating was a locally identified priority, for example by the local government area.

Both government support service providers and the health promotion workforce provided avenues of support for facilities implementing the guidelines; however, overlap in the types of support were identified (“*training and support for implementation*”). Uniquely, HEAS offered programs to build the capacity of health promotion officers and workshops and online training for program implementers (“*capacity building*”), and AP offered support with achieving the “healthy eating and oral health” benchmark. However, almost all support organisations offered one-to-one implementation and menu assessment support, and resources and tools to support implementation (“*provision of information*”). Technical assistance was also offered by many, such as menu improvements, recipe modification and working with suppliers on behalf of organisations (“*technical/troubleshooting assistance for implementation*”). In identifying areas of overlap, the analysis demonstrated a clear opportunity to streamline the support system and consolidate implementation materials, with adequate resourcing of support service providers and the health promotion workforce (Figure 2). In the streamlined system, the focus of HEAS would be on capacity-building of the health promotion workforce (and providing them with support and technical assistance when necessary), allowing them to provide implementation support to implementing organisations. HEAS would continue to provide information and guidance, tools and resources to support statewide implementation.

##### Quality and Utility of Support

Stakeholders discussed the quality and utility of support given by state-funded support services. The types of support and technical assistance provided were well-perceived by most, but some shared that the level of support provided (i.e., the amount of time and depth of support) was inadequate. Seven stakeholders suggested this was due to lack of capacity and resources within support service providers, while support service providers shared that it was “not sustainable to continue hand-holding people but that’s what’s needed” (Support—Workplaces, hospitals and/or sport and recreation). Reliability, such as support services consistently being there to “pick up the phone” (Implementer—Workplace (local government area)/Support—Sport and recreation) when interviewees needed support or assistance was seen by some as an enabler for implementation. Capacity-building of the health promotion workforce by HEAS was identified as a key opportunity for scaling up implementation support across the state. The resources provided by HEAS to support implementation were generally regarded as helpful. However, almost half of the interviewees cited gaps in currently available resources and suggested ideas for resource development, such as a step-by-step implementation guide and a healthy suppliers list. Six interviewees, including two suppliers and four program implementers, stated a major barrier to accessing support was the cost for services, particularly menu and product assessment charges (“*accessibility of support*”).

#### 3.3.3. Delivery System

##### Implementation Process

Participants consistently reported a step-wise approach to implementation, and from the synthesis and analysis of interviews we determined that successful implementation required a specific pathway from start to finish (Figure 3; “*Steps to implementation*”). Although, not all steps were undertaken by all organisations.

The main reasons to implement the Guidelines were identified: achieving recognition/accreditation (i.e., AP), an individual’s motivation for creating healthy food environments, a sense of organisational responsibility for providing healthy foods or a management initiative (“*Motivation to implement*”). Most organisations accessed at least one support service (see Support System). Forming an internal committee/working group with defined individual and group roles and responsibilities was viewed as a key enabler. These committees/working groups often included executives/management, program implementers, supporters within the organisation and retailers/suppliers. Adequate staff resourcing and training were critical to timely and effective implementation. Baseline assessments of food provision methods, licencing agreements/contracts, meals and foods available, and the external food environment were needed to inform the implementation process. Many organisations created their own healthy eating policy that incorporated the HCG or SCFSP and organisational values, which facilitated a sense of ownership. A staged approach to implementation was then commenced, starting with small steps of least resistance and gaining easy wins. Two methods were frequently applied: a food-based approach (e.g., first removing sugary drinks, then moving on to other drinks and subsequently foods) and a food-provision type approach (e.g., starting with vending machines, then moving onto catering and subsequently retail). One organisation undertook both approaches: “So, we started with drinks and we’ve broken down our projects into four of five different domains. The next things we’re working on our vending and catering and then we’ll work on food outlets down the track.” (Implementer—Hospital or Health Service). Overall, the most common implementation strategies reported were: changes to product and variety (*n* = 17), recipe modifications (*n* = 14), simple healthy food swaps (*n* = 11) and strategic placement to promote healthy foods (*n* = 11) (“*Implementation strategies used by organisations*”). Often these strategies were supplemented by consumer education strategies (*n* = 18). Monitoring and evaluation were frequently discussed by interviewees as the final and continual step. This included annual menu auditing (either external or self-evaluation) and surveys about staff/community attitudes (“*Steps to maintain implementation*”). However, most organisations indicated that they “don’t have the resources or the tools to be monitoring and evaluating” (Implementer—University) on an ongoing basis (see General Capacity).

##### Enablers and Barriers Affecting Delivery within Organisations

Enablers and barriers affecting guideline implementation were identified inductively. These were categorised into individual, organisational, community, supplier/retailer and governance factors (Table 3). The primary barrier and enabler for each category are listed in the following paragraphs.

In order, the main barriers included: (1) lack of individual understanding of what healthy eating is, (2) inadequate leadership support and staff resourcing within organisations, (3) resistance from organisational staff and suppliers/retailers, (4) unavailability of healthier commercial foods and challenges in creating healthier options, and (5) lack of governance, enforcement and accountability mechanisms (Table 3). Barriers were often exacerbated by the complexity of the organisational environment, such as multiple sites/venues and multiple methods of food provision, particularly when food provision was outsourced.

The key enablers participants described to overcome these barriers were: (1) involvement of individuals with previous experience and networking with other organisations that had either successfully implemented the guidelines or were also in the process of implementation, (2) receiving upper management/executive support for implementation, (3) having adequate resources (primarily staff time and funding), (4) gaining support from and creating working relationships with staff, customers, suppliers/retailers and other key stakeholders, and (5) building healthy eating into contractual obligations when a contract was up for tender to overcome supplier/retailer resistance (Table 3). Additionally, framing the guidelines to be about increasing the availability of healthier foods and organisational responsibility for providing healthier food options seemed to overcome some concerns about taking away individual choice (i.e., nanny state).

##### General Capacity

Participants described how general capacity was hindered by inadequate staffing and resources at most facilities (“*staffing and resources*”). Lack of time was frequently stated, with many interviewees sharing that they worked on implementing the guidelines outside of work hours as it was not built into anyone’s role. Deficiencies in skills and expertise could be overcome through engaging with the support system (“*skills and expertise*”).

Timeframes for implementation were often considered unrealistic, with many stakeholders sharing that implementation is ongoing after 3–5 years (“*timeframe for implementation*”).

## 4. Discussion

Enablers and barriers to the implementation of government nutrition standards in publicly funded institutions in Victoria were identified through the analysis of perspectives of 44 stakeholders. The Interactive Systems Framework facilitated the identification of factors supporting or hindering implementation from policy design and governance, through the support system to the organisational level. This increased our understanding of opportunities to support, or intervene, to improve statewide implementation progress. Overall, effective statewide implementation of government nutrition standards in publicly funded institutions in Victoria depends on a streamlined, well-resourced support system, improved policy governance and accountability, action from private food companies and addressing misconceptions about the guidelines. These findings have wider implications for the implementation of food policies to create healthy eating environments globally.

### 4.1. A Streamlined and Well-Resourced Implementation Support System Is Required for Effective Statewide Implementation

The existing state-funded implementation support service, HEAS, was viewed as helpful in bridging the policy design and implementation gap. It was clear that without this service, and the resources it provided, many end-users would not be able to understand or apply the guidelines due to their complexity and assumed knowledge. This finding is reinforced by evidence from other jurisdictions around the world, in which a lack of user-friendly information, implementation support and capacity-building has been a key barrier to implementing nutrition standards [31,32] and other food policies, such as menu labelling [33]. In fulfilling the key roles of producing end-user resources, problem-solving through the provision of technical support and troubleshooting, and capacity-building of both the health promotion workforce and staff within implementing organisations [30,34], HEAS plays an essential part in mitigating barriers to the implementation of institutional nutrition standards in Victoria.

Streamlining implementation support pathways through clear allocation of support roles to reduce duplication of effort could overcome barriers to implementation due to insufficient capacity and resourcing. Providing implementation support at scale (in this case, across all publicly funded institutions in the state of Victoria) is notably a challenge. Some studies suggest the impact of implementation support is lessened when health promotion policies and programs are administered en masse, likely due to the need to adapt the types of support able to be provided [35,36]; although others have demonstrated at scale effectiveness so long as key support components were provided [37]. Our analysis highlighted that the government-funded support services in Victoria (HEAS and AP) provide key support elements [37] to publicly funded institutions, including FoodChecker for auditing and AP for recognition, workshops and training for capacity building, and tools and resources for implementation. Generating leadership support and guiding consensus processes (e.g., forming a healthy eating committee and/or creating an organisational healthy eating policy that included the guidelines) were identified by stakeholders as additional activities that HEAS could support to further increase the impact of the service. Overall, the Government-funded statewide support services facilitated guideline implementation and achievement. However, capacity is already a challenge and if demand increases, such as through mandating the guidelines, the streamlining of the support system, alongside increased system resourcing, will be vital for effective implementation.

### 4.2. Improved Policy Governance and Accountability Is Crucial for Effective Statewide Implementation

Program implementers and supporters struggled to generate sufficient leadership and organisational support to commence implementation, as a lack of policy governance meant there was no external motivation to implement or achieve the guidelines. The consequence of this was insufficient staffing and resourcing, leading to a slow and difficult implementation process. A common remark was the lack of a statewide mandate for guideline implementation. Nutrition standards are voluntary in most publicly funded institutions in Victoria, which is likely a function of a lack of public and political will for preventative health policies [38]. Meanwhile, globally, mandatory nutrition standards are becoming increasingly common in publicly funded institutions [3]. This shift likely reflects growing evidence about the effectiveness of regulatory approaches [39] and increasing perspectives that governments have a responsibility to regulate the types of foods that can be procured, prepared, provided or sold using public money [1,40]. A key implication of the voluntary nature of the standards in Victoria is the lack of accountability mechanisms, such as a statewide monitoring system, compared to other jurisdictions (e.g., New South Wales [41]). Accountability measures are crucial for the successful implementation of any nutrition policy [42]. Without systems in place, voluntary guidelines will likely be ineffective, as seen in the implementation of nutrition standards in sport and recreation settings in the Canadian province of Nova Scotia [43]. In Victoria, the effect of not having accountability mechanisms is already apparent with the SCFSP. Independent menu assessments have revealed a lack of compliance: No schools were meeting the minimum compliance criteria in 2008–2009 [44] or 2019 [45], and 40–50% were selling prohibited items [44,45]. Improving policy governance of both HCG and SCFSP will be crucial if the guidelines are to be effective. The 2021 policy directive for hospitals and health services to implement nutrition standards across all food retail, vending and catering by 2023 [46], and the planned establishment of monitoring mechanisms for this, is a key opportunity. Once established, the accountability systems should be applied across all publicly funded institutions, regardless of whether the guidelines are mandatory or voluntary, to stimulate statewide guideline uptake, implementation and compliance.

### 4.3. Systemic Change Is Essential for Effective Statewide Implementation

A step-wise approach to successful implementation was identified through the analysis, illustrating innovative ways publicly funded institutions are overcoming guideline implementation barriers; however, systemic change including actions by government and private food companies will be a prerequisite for institutions across the state to achieve the guidelines. Resistance from private food companies, stemming from perceived business risks, inadequate capacity and skills, and difficulties creating (or accessing) healthier foods, hindered implementation in many organisations. This has also been seen in other samples of Victorian stakeholders [47] as well as studies in Canada, the United States and other countries [31,48,49]. Organisations used one of three approaches to encourage food companies to overcome these challenges: building guideline compliance into new or renewed contracts with suppliers/retailers and monitoring compliance, changing retailers/suppliers to a healthier one, effectively eliminating the need for the organisation to undergo the complex and lengthy guideline implementation process; or establishing relationships with retailers/suppliers and working together to make incremental changes to the healthiness of foods available. Undertaking a collaborative approach was effective in engaging stakeholders and commencing guideline implementation [50,51] but did not lead to complete implementation. The other two approaches can lead to full implementation but depend on opportunities for new or modified contracts when previous ones expire. Contractual obligations are an important policy lever [52], and for effective statewide implementation, all contracts between publicly funded institutions and retailers/suppliers should include a requirement for compliance with the guidelines. This can be enacted at the state or local government level, or in the absence of government action, within organisations. At present, compliance may be a challenge, as a lack of suppliers/retailers providing healthier foods was a key barrier. This demonstrates a key marketing and business opportunity for suppliers and retailers that can provide permitted foods to publicly funded institutions in Victoria, a finding that is relevant to manufacturers and suppliers globally [31]. External actors, including governments and private food companies, will play a key role in overcoming barriers to enable effective statewide implementation.

### 4.4. Addressing Misconceptions Is Necessary for Effective Statewide Implementation

The variety of misconceptions cited by stakeholders likely reflects a broader lack of understanding about the guidelines, the purpose of the guidelines and support services available. A disparity between the guidelines’ definition of healthy food and individuals’ perceptions of healthy food was stated as a major barrier to implementation, as it resulted in a lack of support by key actors [53]. There were evidence-based concerns, such as the focus on reduced-fat dairy products, which was perceived to be out-of-date given the 2019 National Heart Foundation of Australia guidance [54]. The Victorian guidelines are based on the Federal Government’s Australian Dietary Guidelines [55], as are all other institutional nutrition standards in Australia [10]. However, these were last updated in 2013 and a review is currently underway to update them in accordance with the latest evidence [56], which could include the new evidence on reduced-fat dairy products [54]. Institutional nutrition standards across Australia will need to be revised in line with the updated Australian Dietary Guidelines. Other concerns stemmed from personal beliefs about what constitutes an unhealthy product (such as ultra-processed foods, the use of additives and preservatives, and artificial colours and flavours) and misconceptions stemming from fad diets (such as replacing all fats and oils with coconut oil, or trying to create a Paleo or sugar-free school canteen). It is already known that individuals’ misconceptions of healthy foods influence their personal food choices [57] and usually result in poorer individual diet quality [58]. From this research, it appears these misconceptions can also hinder the implementation of evidence-based healthy food guidelines in publicly funded institutions, due to a lack of personal support for implementation.

The purpose of the guidelines was another area of contention. Interviewees shared that some actors within their organisation opposed the guidelines because they perceived the guidelines controlled or took away individual food choices. Others stated the guidelines were intended to create healthy eating environments and reflected organisational responsibility for providing and promoting healthier food options. This tension between government and organisational responsibility for health-promoting policies versus personal choice and responsibility is frequently discussed in the literature [59]. This can be a major barrier to policy implementation, as illustrated previously in the implementation of nutrition standards in sport and recreation centres and schools in Canada [49,60]. Stakeholders highlighted that focusing on increasing the number and type of healthy options available while providing a limited amount and less variety of unhealthy options was important for balancing this tension and pursuing guideline implementation.

Finally, six interviewees stated a major barrier to accessing the state-funded support service, HEAS, was the cost of services. However, HEAS is a free government-funded support service for publicly funded institutions in Victoria [61]. It is possible these participants used Nutrition Australia consultancy services for a fee, which are supplementary to HEAS free services [62]. Organisations may be utilising paid services from Nutrition Australia due to insufficient capacity at HEAS to meet the organisations’ support service needs. They may also be paying for Nutrition Australia to undertake implementation activities, such as menu assessments, because the organisation does not have the capacity to complete these themselves using HEAS support or resources. Since this is a unique service, the effect of this perceived cost on service access, and hence implementation, is unknown. It is crucial that organisations seek and receive clarity around which service is being accessed and if there is an associated cost. Together, these findings around stakeholder misconceptions point to the need for stronger communication from the State Government and support services to publicly funded institutions about the evidence informing the guidelines, the purpose of the guidelines and the free implementation support available.

### 4.5. Summary of Lessons for the Implementation of Nutrition Standards in Publicly Funded Institutions

Taken together, the findings of this study and global evidence suggest that effective implementation of nutrition standards in publicly funded institutions requires strong governance and accountability, easy-to-understand guidelines, adequate support from external providers and a step-wise approach to implementation.

Regardless of whether policies are voluntary or mandatory, compulsory accountability mechanisms (including ongoing monitoring and evaluation procedures) must be in place to (i) ensure the government, manufacturers/suppliers to institutions, and institutions are answerable to the public, and (ii) enable statewide/national uptake, implementation and compliance by stimulating sufficient organisational leadership and support for implementation. To guarantee supplier/retailer compliance, contracts between publicly funded institutions and private retailers/suppliers, which are an important lever for effective implementation, should include a requirement for adherence to the guidelines.

Victoria is unique in that there exists Government-funded services to support the implementation of nutrition standards in publicly funded institutions. This research highlights how support services can help organisations implement nutrition standards and overcome implementation barriers. In other contexts, international and local non-government organisations can play a similar role.

As our previous research has highlighted [3,10], nutrition standards must be easy to understand and implement. This study reinforces that long and complex nutrition guidelines are a barrier to effective implementation as misconceptions about the guidelines can arise, and it is challenging to achieve and monitor compliance.

This study found that effective implementation in Victoria requires a series of steps. There is considerable commonality between these steps and the new WHO Action Framework [40], which can be used by countries globally to develop and implement nutrition standards in publicly funded institutions. 

### 4.6. Strengths and Limitations

Strengths of this study included the development of a purpose-built pre-interview questionnaire and survey instrument, which was based on a well-used and evaluated framework aiming to understand the gap between policy design and implementation. The framework was also used to guide the analysis, which was expanded upon by the authors using an inductive approach. However, it should be noted that others are moving towards a complex systems approach [47], which may be useful for future researchers. Interviews were conducted with a wide variety of stakeholders, who had different roles and worked in different publicly funded institutions, enabling a broad range of perspectives. The semi-structured nature of the interviews facilitated a deeper understanding of these diverse perspectives. However, in undertaking this approach, there were only a few participants representing some stakeholder groups. Specifically, due to the low number of interviewees who supported the implementation of HCGs across workplaces, hospitals and sport and recreation centres, we analysed these stakeholders’ perspectives together as one group. These stakeholders had similar experiences and perspectives and it is unlikely this grouping influenced the results. In addition, the recruitment approach, which was through the support service mailing list, was chosen due to previously faced challenges in contacting and engaging eligible participants within publicly funded institutions as part of a larger statewide salt reduction program [63]. However, our sample was likely biased towards organisations implementing or already complying with the guidelines, so perspectives may not be representative of all stakeholders in publicly funded institutions in Victoria.

## 5. Conclusions

This study has identified opportunities to improve the design of nutrition standards and close the policy-implementation gap to maximise the impact of nutrition standards on population diets and reduce diet-related disease. This study provides insight into the usefulness of a comprehensive support system, including government-funded support services, in facilitating the implementation of nutrition standards in publicly funded institutions, as well as prospects for further improvement. Strengthening the guidelines and their governance, streamlining the support system and overcoming barriers within and outside of implementing organisations are urgently required to propel statewide progress.

## Figures and Tables

**Figure 1 nutrients-14-02628-f001:**
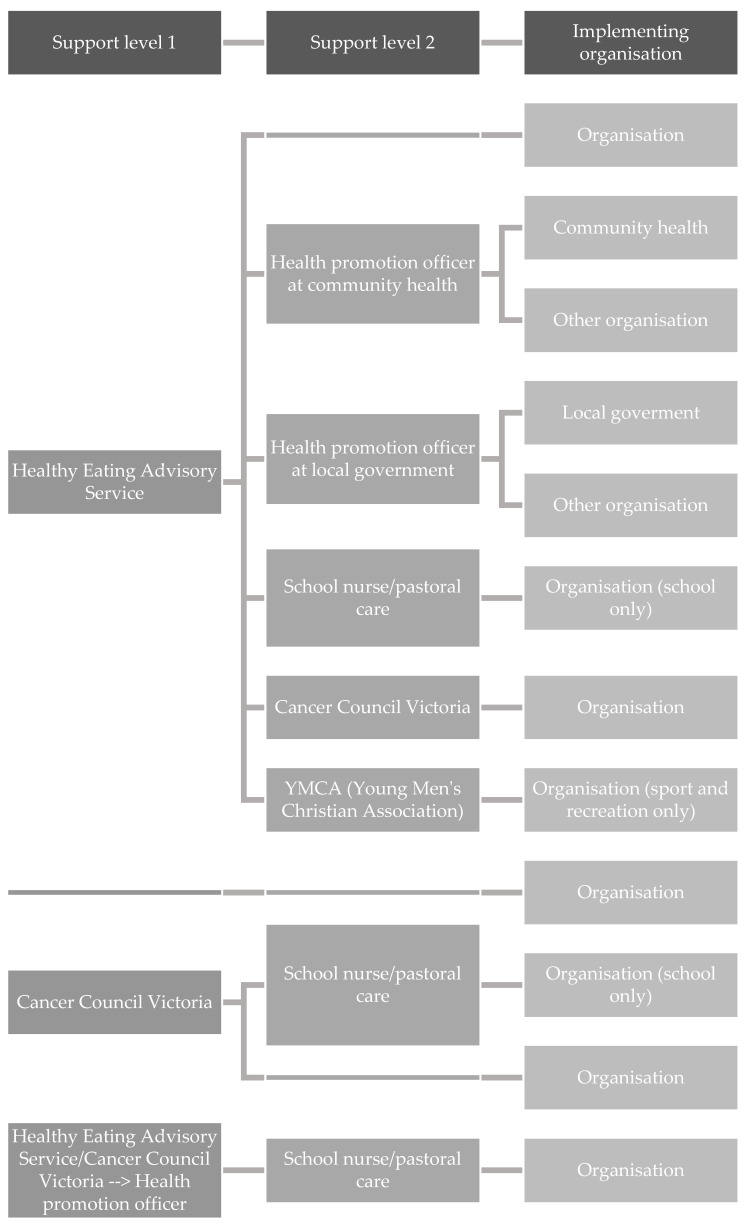
Current pathways of support for facilities implementing Healthy Choices Guidelines and School Canteens and Other Food Services Policy identified from the interviews.

**Figure 2 nutrients-14-02628-f002:**
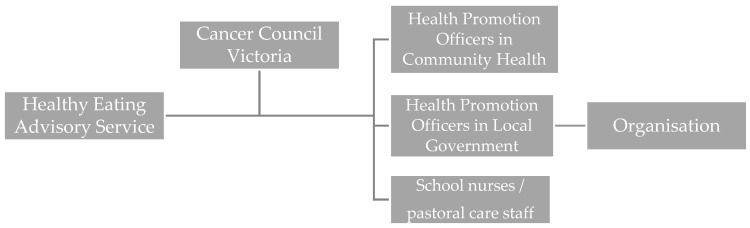
Streamlined support pathway to implement the Healthy Choices Guidelines and School Canteens and Other Food Services Policy determined from the analysis of the interviews.

**Figure 3 nutrients-14-02628-f003:**
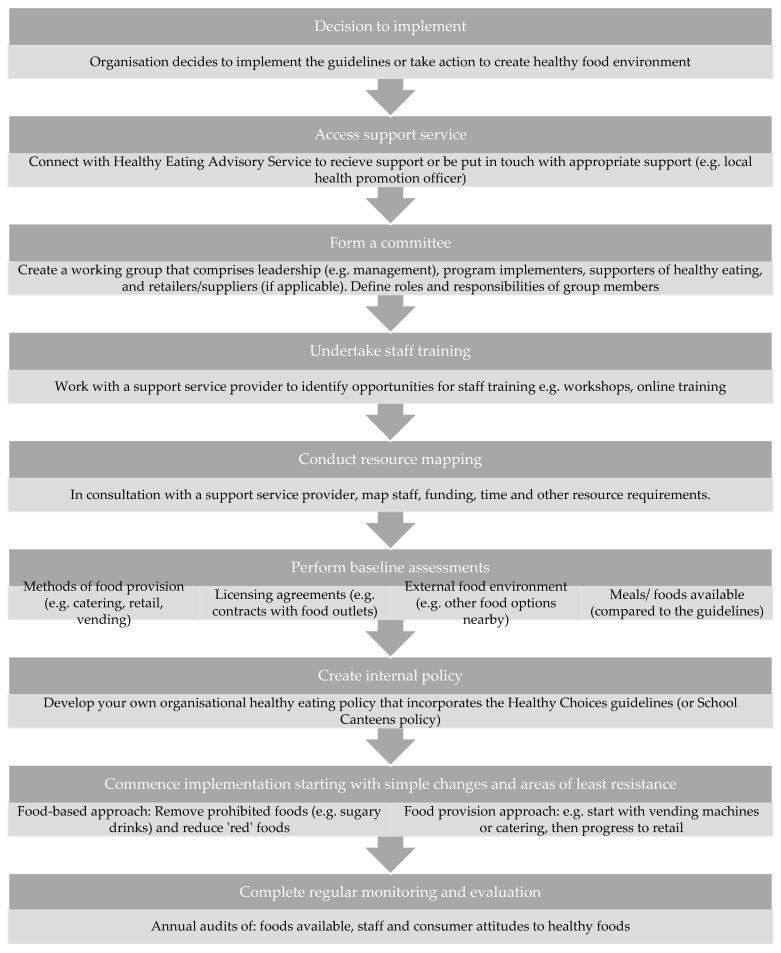
Recommended steps to implementing the guidelines determined through the synthesis and analysis of interviews.

**Table 1 nutrients-14-02628-t001:** Adapted Interactive Systems Framework.

System	Constructs and Sub-Constructs ^1^
Synthesis and translation system	**Accessibility**
	**Understandability**
	**Utility**
Support system	***Information and support*** Provision of information
	Training and support for implementation
	Technical/troubleshooting assistance for implementation
	Capacity building
	***Quality and utility of support*** Accessibility of support ***Structure of the support system***
	*Relationship between different support services*
	*Relationship between support services and organisations*
Delivery system	***Implementation process*** Motivation to implement (e.g., accreditation)
	*Steps to implementation (e.g., staged approach to change)*
	*Implementation strategies used by organisations (e.g., recipe modifications)*
	*Steps to maintain implementation (e.g., annual audits)*
	** *Enablers and barriers affecting delivery within organisations* **
	*Individual factors (e.g., previous experience in implementation)*
	*Organisational factors (e.g., leadership engagement/support)*
	*Supplier factors (e.g., contractual issues)*
	*Community factors (e.g., consumer demand)*
	*Governance factors (e.g., lack of enforcement)*
	**General capacity**
	Staffing and resources (e.g., adequate staff)
	Skills and expertise (e.g., nutrition skills)
	Timeline for implementation

^1^ Bold illustrates main constructs; italics indicates inductively identified constructs.

**Table 2 nutrients-14-02628-t002:** Number of participants by role in guideline implementation.

Role	Number of Interviewees
Program implementer—Hospital or health service	14
Program implementer—School	10
Program implementer—Workplace (local government area)/Support—Sport and recreation	4
Program implementer—Workplace (university, government facility)	4
Support—Workplaces, hospitals and/or sport and recreation	5
Support—School	2
Manufacturer/Supplier	5

**Table 3 nutrients-14-02628-t003:** Key enablers and barriers influencing implementation.

Key Factors	Enablers	Barriers
Individual	- Previous experience in a similar role - Networking with other organisations	- Disparity between individuals understanding of what is “healthy” and the policy/guidelines
Organisational	- Upper management support - Leadership engagement - Paid staff - Adequate resources	- Lack of upper management support - Lack of leadership engagement - Inadequate resources (particularly volunteer-run organisations) - Complex food environment
Community	- Staff buy-in/support - Customer demand for healthier foods	- Staff backlash/resistance - Customer opposition
Supplier/retailer	- Building a positive relationship with suppliers/retailers - Supplier/retailer motivated to provide healthier food	- Resistance from supplier/retailer - Having to change suppliers/retailers - Reliance on supplier-provided fridges and equipment - Fear of profit loss - Healthier commercial foods not available
Governance	- Building policy/guidelines into contractual obligations with suppliers and retailers	- Voluntary guidelines - Lack of enforcement

## Data Availability

The data presented in this study are available on request from the corresponding author. The data are not publicly available due to ethical reasons.

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
