# Peer review of "Understanding Enablers and Barriers to the Implementation of Nutrition Standards in Publicly Funded Institutions in Victoria"

_nutrients, 2022, doi:10.3390/nu14132628_

Round 1

Reviewer 1 Report

Thank you for the opportunity to review this paper. This study mainly explored stakeholder perspectives on the enablers and barriers to successfully implementing government nutrition standards in publicly funded institutions in the Australian state of Victoria. It is an interesting and important topic, and the authors made great efforts to complete the study. I have a few comments to improve this manuscript. 

Line 39: Please list a few examples of publicly-funded institutions such as what you have mentioned in lines 51-52.

Are the few nutrition policies listed in the third paragraph all policies or just a few examples? If examples, please add something as “For example” at the beginning of line 56. 

Line 82: Please define YMCA when you first mention it. 

Lines 109-111: Are potential participants and eligible interviewees the same? A little confusing here. 

Line 126: I suggest deleting the sentence “At this time, 13 interviews had been conducted.” Since you have not mentioned the number of interviewees here. It seems that you conducted 44 interviews in total: why only 13 here? 

Line 146: Are the supplementary materials the same as your supplementary table 1 in line 206?

Line 158: is 10% of transcripts good enough to get unbiased results? Is there any standard for this? How did you resolve disagreements in coding? 

I believe the entire table 1 is part of the results, so please refer to the table first in the result section. 

Line 199: n=38 organizations? You mentioned there were 36 organizations in total?

It seems that your entire section 3.3 was explained based on the themes you summarized in table 1. But it is hard to find out the points you are emphasizing in 3.3 based on table 1. It may be helpful to divide sections based on these themes or at least make them clearer. 

Is Figure 1 an existing figure, or did you design it based on the interview results? Need to be more explicit, same as figure 2 and figure 3. Give more details about how the figures were generated. 

You named section 3.3 as “Enablers and barriers to implementation.” Then in 3.3.3.1, you clearly described the enablers and barriers affecting delivery using a subsection, but you did not do the same for 3.3.1 synthesis and translation and 3.3.2 support, where it was hard to find enablers or barriers. 

Some of the content in the discussion section lacks connection with the results. 

This study collected a great amount of valuable information to promote state-wide nutrition policy implementation. But the authors need to consider rearranging some content to make the structure clearer, so the readers do not need to check back and forth to understand this important study. 

Author Response

Thanks for your review. Please see attached point by point response. 

Reviewer 2 Report

Thank you for the opportunity to review this important manuscript, which I believe will be good, with some minor revisions. This manuscript has potential to make an important contribution to the literature on nutrition policy implementation, including facilitators and barriers. I have provided some specific comments, as follows.

Literature Review: Please add 2-3 sentences about the broader literature on the evaluation of nutrition policies. As it currently stands, that piece seems omitted, but would  be helpful in setting the broader context for this research.

Methods: You administered a pre-interview questionnaire, and then tailored your interview guide/questions, and adapted for the interviewee. My question is about how you analyzed the interview data in the same way if the questions were different?

Line 126: You mentioned that thirteen interviews were conducted about HGC by March 2020; yet in line 178, you state that 44 stakeholders had been interviewed. Can you please include a clarifying statement here about how many interviews were conducted, and when, as in its current state, it is confusing for the reader?

Discussion section: how are these findings potentially relevant or transferrable to similar or different contexts where nutrition policies are implemented? What about in other countries with similar policies?

Author Response

(The authors gave the same response as above.)
